# Effects of Mycotoxin Fumagillin, Mevastatin, Radicicol, and Wortmannin on Photosynthesis of *Chlamydomonas reinhardtii*

**DOI:** 10.3390/plants12030665

**Published:** 2023-02-02

**Authors:** Jiale Shi, Mengyun Jiang, He Wang, Zhi Luo, Yanjing Guo, Ying Chen, Xiaoxi Zhao, Sheng Qiang, Reto Jörg Strasser, Hazem M. Kalaji, Shiguo Chen

**Affiliations:** 1Weed Research Laboratory, Nanjing Agricultural University, Nanjing 210095, China; 2Bioenergetics Laboratory, University of Geneva, CH-1254 Jussy, Geneva, Switzerland; 3Department of Plant Physiology, Institute of Biology, Warsaw University of Life Sciences SGGW, Nowoursynowska 159, 02-776 Warsaw, Poland

**Keywords:** natural product, photosynthetic inhibitor, mode of action, JIP-test, molecular docking, D1 protein

## Abstract

Mycotoxins are one of the most important sources for the discovery of new pesticides and drugs because of their chemical structural diversity and fascinating bioactivity as well as unique novel targets. Here, the effects of four mycotoxins, fumagillin, mevastatin, radicicol, and wortmannin, on photosynthesis were investigated to identify their precise sites of action on the photosynthetic apparatus of *Chlamydomonas reinhardtii*. Our results showed that these four mycotoxins have multiple targets, acting mainly on photosystem II (PSII). Their mode of action is similar to that of diuron, inhibiting electron flow beyond the primary quinone electron acceptor (Q_A_) by binding to the secondary quinone electron acceptor (Q_B_) site of the D1 protein, thereby affecting photosynthesis. The results of PSII oxygen evolution rate and chlorophyll (Chl) *a* fluorescence imaging suggested that fumagillin strongly inhibited overall PSII activity; the other three toxins also exhibited a negative influence at the high concentration. Chl *a* fluorescence kinetics and the JIP test showed that the inhibition of electron transport beyond Q_A_ was the most significant feature of the four mycotoxins. Fumagillin decreased the rate of O_2_ evolution by interrupting electron transfer on the PSII acceptor side, and had multiple negative effects on the primary photochemical reaction and PSII antenna size. Mevastatin caused a decrease in photosynthetic activity, mainly due to the inhibition of electron transport. Both radicicol and wortmannin decreased photosynthetic efficiency, mainly by inhibiting the electron transport efficiency of the PSII acceptor side and the activity of the PSII reaction centers. In addition, radicicol reduced the primary photochemical reaction efficiency and antenna size. The simulated molecular model of the four mycotoxins’ binding to *C. reinhardtii* D1 protein indicated that the residue D1-Phe265 is their common site at the Q_B_ site. This is a novel target site different from those of commercial PSII herbicides. Thus, the interesting effects of the four mycotoxins on PSII suggested that they provide new ideas for the design of novel and efficient herbicide molecules.

## 1. Introduction

Mycotoxins are low-molecular-weight secondary metabolites naturally produced by fungi. These natural products are one of the most important resource repositories for the discovery of new drugs and pesticides because they possess abundant sources, diverse chemical structures, and wide bioactivity. Generally, they exhibit several phytotoxic, cytotoxic, antimicrobial, and even antitumor activities [1,2]. Some molecular target sites of the known mycotoxins have been described in plants, being involved in amino acid synthesis, energy transfer, PSI electron diverter, PSII electron transport, photosynthetic pigment synthesis, membrane functions and lipid stability, hormonal regulation, and cell cycle [3]. However, the action targets of many mycotoxins in plant cells remain unknown. Therefore, it is important to clarify the mechanism of action of various mycotoxins in plants. This will provide powerful tools for the study of plant physiological and biochemical mechanisms and will promote the development of new herbicides by directly using mycotoxins or new derivatives synthesized based on these lead templates.

Fumagillin (CAS No. 23110-15-8) is a natural biomolecule originally isolated from *Aspergillus fumigatus*. It is a meroterpenoid produced by the esterification of fumagillol and 2,4,6,8-decatetraenedioic acid (Figure 1A) [4]. Fumagillin can inhibit well-functioning proteins related to cell viability and growth, such as methionine aminopeptidase type 2 enzyme [5]. In agriculture, fumagillin is applied as an antibiotic to prevent microsporidiosis in honeybees and fish [6]. Additionally, Ross et al. [7] showed that fumagillin strongly reduces root development due to the inhibition of methionine aminopeptidase 2 in *Arabidopsis*.

Mevastatin (CAS No. 73573-88-3), also called compactin, was first discovered in *Penicillium citinium* [8]. It is a carboxylic ester, consisting of three major functional components, six-membered oxyheterocycle, hexahydronaphthalene, and methylbutyric acid (Figure 1B). Mevastatin is a cholesterol-lowering agent that works by inhibiting 3-hydroxy-3-methylglutaryl-CoA reductase [8]. Thus, it is regarded as a cell apoptosis inducer and has served as a lead compound for the design of new derivatives. Soto et al. [9] suggested that mevastatin can also interrupt the key step of the mevalonate pathway in isoprenoid biosynthesis in plant cytosols.

Radicicol (CAS No. 12772-57-5), known as monorden, was first found in *Monosporium bonorden* [10]. It is also secreted by *Neonectria radicicolas*, *Colletotrichum graminicola,* and other organisms. Radicicol is a 14-membered *β*-resorcylic acid lactone with a 17R configuration, which is characterized by the resorcyclic acid moiety replaced by a chlorine atom (Figure 1C). It shows significant antibacterial, antifungal, antiviral, anticancer, and antiparasitic activities [11]. Recently, it was demonstrated that radicicol is a specific inhibitor of heat shock protein 90 ATPase activity in *Chlamydomonas reinhardtii* and *Arabidopsis*, reversibly inhibiting the import of abundant preproteins during transmembrane transport [12].

Wortmannin (CAS No. 19545-26-7) is a furanosteroid metabolite derived from the endophytic fungi *Fusarium oxysporum*, *P. wortmannii*, and *P. funiculosum* [13]. It is an organic heteropentacyclic compound containing *δ*-lactone, acetate ester, and cyclic ketone (Figure 1D). Wortmannin is a dose-dependent inhibitor of phosphatidylinositol 3-kinase and phosphatidylinositol 4-kinase [14]. Therefore, it is an extremely helpful tool in the disruption and identification of vesicular trafficking routes and the definition of endosomal compartments [15]. Wortmannin exhibits excellent cytotoxic activity against human cancer cells [16]. In plant cells, wortmannin interferes with vacuolar transport and causes homotypic fusion and enlargement of multivesicular bodies [17]. It can also accelerate chloroplast division by repressing phosphatidylinositol 4-kinase activity [18].

Photosynthesis is the most important metabolic process in plant growth and development, being very sensitive to changes in environmental conditions. Photosystem II (PSII), as one core of photosynthetic reactions, is a large protein complex located in the thylakoid membranes, including the D1 and D2 proteins as well as several cofactors [19]. Because photosynthesis is specific to green plants, it has attracted much interest in the development of chemical herbicides. Over 50% of the commercial herbicides on the market are photosynthetic inhibitors. Furthermore, fifteen primary targets are located in the photosynthetic apparatus chloroplast among the less than thirty different molecular targets of commercial herbicides [20]. Thus, photosynthesis is often considered a priority when the precise target of a new herbicidal compound needs to be identified in plants. However, up to now, there has been little information on the targets of fumagillin, mevastatin, radicicol, and wortmannin in the photosynthetic apparatus.

In this study, we hypothesized that these four mycotoxins negatively affect photosynthesis, especially PSII activity. To test this hypothesis, the rate of oxygen evolution and chlorophyll (Chl) *a* fluorescence signal of *C. reinhardtii* cells treated with four different mycotoxins were measured to identify their sites of action in photosystems. Because *C. reinhardtii* is one of the most widely used model systems for molecular biological studies of photosystems including extensive characterization of its PSII reaction center, it was used in this study. The model of the interaction between each mycotoxin and the D1 protein of *C. reinhardtii* was also constructed by molecular docking to further confirm their possible binding sites in the D1 protein. Finally, it was demonstrated that four mycotoxins have direct effects on PSII and different binding behavior with the D1 protein from the classical herbicide diuron (DCMU).

## 2. Results and Discussion

### 2.1. Effects of Four Mycotoxins on the Oxygen Evolution Rate of PSII

As PSII is an important photosynthetic component that catalyzes light-driven water oxidation, we measured the effect of four mycotoxins on the PSII oxygen evolution rate of *C. reinhardtii* cells. A concentration-dependent decrease in the rate of oxygen evolution of cells exposed to mycotoxin was observed (Figure 2). Fumagillin reduced the rate of oxygen evolution at lower concentrations than the other three mycotoxins. An approximately 61% decrease in the PSII oxygen evolution rate was found in 100 μM fumagillin-treated cells relative to the mock (Figure 2A). However, in the presence of 1000 μM mevastatin, radicicol, or wortmannin, the oxygen evolution rate of *C. reinhardtii* cells was decreased by 51%, 40%, and 45% compared with that of the mock, respectively (Figure 2B,D). All four mycotoxin treatments caused a concentration-dependent decrease in the oxygen evolution rate of PSII, indicating that the photosynthetic activity of PSII was inhibited. In addition, fumagillin was a stronger inhibitor of photosynthetic oxygen evolution than mevastatin, radicicol, and wortmannin. The inactivation of the mycotoxin-induced oxygen evolution mechanism is likely to result in the compromise of its closely related electron transfer efficiency.

### 2.2. Effects of Four Mycotoxins on Chl a Fluorescence Imaging

Chl *a* fluorescence imaging is an expeditious tool used to quickly confirm the photosynthetic damage of green organs, tissues, and even living cells. To further investigate the effect of the four mycotoxins on photosynthetic activity, the Chl fluorescence of mycotoxin-treated *C. reinhardtii* cells was monitored. Based on the results in Figure 2, for subsequent experiments, we chose the concentration of each of the four mycotoxins that had a moderate inhibitory effect on the rate of PSII oxygen precipitation. In Figure 3A, the maximum quantum yields of PSII (F_V_/F_M_) of fumagillin-, mevastatin-, and radicicol-treated *C. reinhardtii* cell shows a gradual decrease with the color fading from blue to dark green. For wortmannin, no obvious effect on the color-coded images of F_V_/F_M_ was observed at the highest concentration of 600 µM. In the case of the highest concentration of fumagillin (50 µM), mevastatin (200 µM), and radicicol (500 µM), the value of F_V_/F_M_ decreased by approximately 57%, 58%, and 59% compared with that of the mock, respectively. A nearly 5% decrease in the F_V_/F_M_ was observed in the presence of 600 µM wortmannin (Figure 3B). The change in the F_V_/F_M_ was consistent with the change in the fluorescence image presented in Figure 3A. Fumagillin was the most effective in suppressing photosynthetic efficiency, followed by mevastatin and radicicol; the suppressive effect of wortmannin was not reflected by the F_V_/F_M_ parameter. The electron transport rate (ETR) was also measured to evaluate the effect of these mycotoxins on photosynthetic activity (Figure 3C). It could be seen that ETR is a more sensitive fluorescence parameter to these four mycotoxins than F_V_/F_M_. After *C. reinhardtii* cells were treated with 40 μM fumagillin, 200 μM mevastatin, 400 μM radicicol, or 400 μM wortmannin, the ETR values decreased to zero. The decrease in the ETR value verified our inference that the electron transfer efficiency was influenced by the mycotoxins. These results suggested that these four mycotoxins should inhibit photosynthetic activity; moreover, fumagillin exhibited a stronger inhibitory activity on photosynthetic electron transport than mevastatin, radicicol, and wortmannin.

### 2.3. Effects of Four Mycotoxins on Chl a Fluorescence Rise Kinetics OJIP

The Chl *a* fluorescence rise kinetics OJIP is an excellent tool to determine the precise effect of different stresses on the photosynthetic apparatus [21]. To investigate the mechanism of action of the four mycotoxins on PSII, the fluorescence rise OJIP curves of *C. reinhardtii* cells were measured after treatment with one of the four mycotoxins for 3 h (Figure 4). It is clear that the raw OJIP curves of DCMU as a positive control and those of each of the mycotoxin-treated cells showed a visible change compared with the typical polyphasic OJIP curve of the mock. Under 50 μM DCMU treatment, the J step quickly increased to the equal P level (F_M_), which is the main change in the OJIP curve relative to the mock. Radicicol and wortmannin at 500 μM, as with DCMU, caused a remarkably fast rise in the J step. Interestingly, the IP phase of the OJIP curve of DCMU- and wortmannin-treated cells disappeared entirely, but still remained in the radicicol-treated cells (Figure 4A). Meanwhile, just a slight increase in the J-step level of the OJIP curve could be observed in 50 μM fumagillin- or 200 μM mevastatin-treated cells. Because a rise in the J step contributes to the large accumulation of reduced primary quinone electron acceptor (Q_A_^−^) in PSII reaction centers (RCs), resulting from the interruption of electron flow beyond Q_A_ at the PSII acceptor side [21,22], it was suggested that electron transferring beyond Q_A_ at the acceptor side of PSII was strongly inhibited by radicicol and wortmannin. This agreed with the ETR results (Figure 3C). However, 50 μM fumagillin and 200 μM mevastatin treatments led to a slight decrease in PSII electron flow beyond Q_A_. In addition, a distinct lift in the O step in fumagillin- and radicicol-treated cells, and an obvious decrease in the P-step level in radicicol-treated cells, were observed (Figure 4A).

To visualize the features that were hidden behind the raw fluorescence rise kinetics OJIP curves, the OJIP curves of mock-, DCMU-, and mycotoxins-treated cells were double -normalized between F_O_ and F_M_, which is expressed as the relative variable fluorescence V_t_ = (F_t_ − F_O_)/(F_M_ − F_O_) (Figure 4B) and ΔV_t_ = V_t(treatment)_ − V_t(mock)_ (Figure 4C) on a logarithmic time scale. It was revealed that the biggest change in the OJIP curves of the mycotoxins-treated cells was a noticeable increase in the J band, which was similar to the effect of DCMU. So, the dominant effect of the four mycotoxins on the photosynthesis of *C. reinhardtii* is blocking PSII electron transfer beyond Q_A_ at the acceptor side.

The differences in the curve changes caused by different mycotoxins are also very interesting. For fumagillin, the O step of the treated cells was slightly elevated, which was similar to the results of DCMU treatment. A rise in the O step represents an increase in the initial yield of fluorescence, which may be the result of the partial or complete inactivity of the oxygen-evolving complexes (OECs) of *C. reinhardtii* cells [23,24]. For radicicol, the O step was significantly higher, whereas the P step was significantly lower. The decrease in the P step is due to fluorescence quenching caused by the interaction between oxidized plastoquinone (PQ) molecules and the PSII antenna [24]. For mevastatin, only a slight increase in the J peak and a slight decrease in the IP phase were observed (Figure 4A). It is normally considered that the I step and IP phase represent the redox state of PQ and the redox state of the terminal receptor on the electron acceptor side of PSI, respectively [21,25]. This suggested that the inhibition of photosynthetic efficiency by mevastatin was mainly attributed to the inhibition of PSII electron transport beyond Q_A_, and the redox status of downstream acceptors was consequently affected. For 500 μM wortmannin treatment, the J peak sharply increased and was similar to the J-step increase for radicicol at the same concentration (Figure 4B,C), indicating that wortmannin and radicicol had similar abilities to inhibit electron transfer, which supports the results in Figure 3C.

### 2.4. Effects of Four Mycotoxins on the Selected JIP Test Parameters

The JIP test parameters give adequate information about the structure, conformation, and function of the photosynthetic apparatus, especially PSII [21]. Some selected JIP test parameters are presented to further confirm the effect of the four mycotoxins on PSII of *C. reinhardtii* (Figure 5). Both fumagillin and radicicol increased the F_O_. Only radicicol clearly decreased the F_M_ value (Figure 5A). Fumagillin, radicicol, and wortmannin caused a decrease in the F_V_/F_M_ which was related to the change in F_O_ and/or F_M_. Mevastatin had nearly no effect on F_O_, F_M_, or F_V_/F_M_. F_O_ and F_M_ are the results of O-step and P-step quantization in the OJIP curve, respectively. The results suggested that fumagillin and radicicol might inhibit the activity of OECs in *C. reinhardtii* cells (increase in F_O_); radicicol might lead to oxidation of the PQ pool or disruption of PSII antennae (decrease in F_M_); and fumagillin, radicicol, and wortmannin have different degrees of inhibition of the quantum efficiency of light energy transfer in PSII (decrease in F_V_/F_M_). In addition, 200 μM mevastatin showed insensitivity to F_V_/F_M_.

All four mycotoxins significantly changed the relative variable fluorescence at the J step (V_J_) and ΔV_J_ (Figure 5B). The changes in these technical fluorescence parameters are perfectly agreement with the results in Figure 4. In the DCMU-treated samples, V_J_ and ΔV_J_ significantly increased due to the large accumulation of Q_A_^−^ in PSII RCs, resulting in the blocking of electron flow beyond Q_A_ [21]. For the treatment with fumagillin (50 μM) or mevastatin (200 μM), the ΔV_J_ values were significantly lower than those of the DCMU (50 μM) treatment. In contrast, the ΔV_J_ values under radicicol (500 μM) or wortmannin (500 μM) treatment were close to those of the DCMU (50 μM) treatment. Thus, the four mycotoxins showed similarities to DCMU in the inhibition pattern of electron transport, but the inhibition effects were significantly different. The inhibition by the four mycotoxins was significantly weaker than that of DCMU, and the inhibition of wortmannin was slightly stronger than that of radicicol. Moreover, radicicol and wortmannin showed an increase in relative variable fluorescence at the I step (V_I_). The values of both fluorescence parameters V_J_ and V_I_ are considered to correlate with the state of electron transport on the PSII acceptor side [21,26], particularly the increases of V_J_ values indicated that the four mycotoxins blocked PSII electron transport beyond Q_A_.

To further verify the inhibition of electron transport from Q_A_ to the secondary quinone electron acceptor (Q_B_) by the four mycotoxins, the parameters φ_Eo_, ψ_Εo_, and ET_0_/RC were analyzed (Figure 5C). φ_Eo_ expresses the quantum yield for electron transport [26]. ψ_Eo_ is the probability that an electron moves further than Q_A_^−^ into the electron transport chain [27]. ET_0_/RC reflects the electron transport per RC [26]. In the presence of the four mycotoxins, the values of φ_Eo_, ψ_Εo_, and ET_0_/RC were all significantly decreased relative to mock, indicating that the four mycotoxins did interrupt PSII electron transfer from Q_A_ to Q_B_.

Because of the negative effect of the four mycotoxins on PSII electron transport, it as expected that the PSII RCs’ activity might have been inhibited as a result. The values of S_m_/t_FM_ in radicicol- and wortmannin-treated cells were clearly lower than that of the mock (Figure 5D). The parameter S_m_ denotes the normalized total complementary region above the O-J-I-P transient, reflecting multiple-turnover Q_A_ reduction events [27]. The parameter t_FM_ means the time to reach F_M_ [26]. The S_m_/t_FM_ ratio represents a measure of the average redox state of Q_A_^−^/Q_A_ from zero to t_FM_ time and expresses the average fraction of open RCs [28]. However, the values of S_m_/t_FM_ did not significantly change in fumagillin- and mevastatin-treated cells, indicating that fumagillin (50 μM) and mevastatin (200 μM) probably do not affect the activity of PSII RCs. Another fluorescence parameter, R_J_, explains the inactivation of PSII RCs. It is generally considered that R_J_ reflects the number of PSII RCs with the Q_B_ site filled populated by PSII inhibitors [27]. For fumagillin- (50 μM), mevastatin- (200 μM), radicicol- (500 μM), and wortmannin- (500 μM) treated cells, the values of R_J_ increased by 16%, 13%, 74%, and 84% relative to the mock, respectively (Figure 5D). This suggested that the four mycotoxins inhibited PSII electron transfer activity due to their filling in the Q_B_ site. Thus, we suggest that the inactivation of the PSII RCs caused by radicicol and wortmannin is attributable to the inhibition of PSII electron transport. The insignificant effect of fumagillin and mevastatin on PSII RCs may be owing to the insufficient number of PSII RCs at the Q_B_ site filled by them and thus the insufficient inhibition of electron transfer.

The values of the N and S_m_ parameters further support the above results (Figure 5E). The turnover number N represents the number of Q_A_ reduction events between time zero and t_FM_ [26]. Compared with the mock, the values of N and S_m_ slightly decreased for fumagillin and mevastatin treatment and sharply decreased for radicicol and wortmannin treatment. Wortmannin treatment especially decreased the N and S_m_ values almost to zero. The changes in these JIP test parameters with wortmannin-treated cells were much stronger than those of the other mycotoxins, which as almost equivalent to that of DCMU (Figure 5C,E). A decrease in S_m_ represents a reduction in the total electron-accepting capacity of leaves, which measures the electron transporter pool between PSII and PSI acceptors [29,30]. Wortmannin strongly blocked electron transport and thus caused a severe inactivation of PSII RCs. So far, there are many reports of the inactivation of PSII RCs caused by natural products with PSII inhibitory activity. It was reported that fischerellin A [31] significantly increased the J step in cyanobacteria, green algae, and pea leaves, inhibiting electron transport and leading to the inactivation of PSII RCs. Tenuazonic acid [27] and patulin [32] have been found to inhibit electron flow from Q_A_ to Q_B_ in higher plants as well, thereby inactivating PSII RCs. Previous studies have shown that a decrease in the number of active PSII RCs causes an increase in non-Q_A_-reducing centers [21,27]. The non-Q_A_-reducing centers, namely the heat sink centers, eliminate the redundant excitation energy mainly by increasing thermal dissipation, leading to harmful reactive oxygen species (ROS) generation.

PI_ABS_ is the most sensitive JIP test parameter, which is used to assess the photosynthetic performance indexes of samples [27]. When *C. reinhardtii* cells were incubated with fumagillin, mevastatin, radicicol, or wortmannin, the values of PI_ABS_ decreased by 61%, 31%, 95%, and 93% compared with that of the mock, respectively (Figure 5F). Significant changes in parameter PI_ABS_ are closely related to the antenna size (γ_RC_), the maximum quantum yield of primary photochemistry (φ_Po_), and the probability of electrons on the reduced Q_A_. further entering the electron transfer chain [21,27]. The treatment with fumagillin and radicicol reduced γ_RC_ by 15% and 35%, respectively. The φ_Po_ of *C. reinhardtii* cells decreased by 18%, 37%, and 13%, respectively, under fumagillin, radicicol, and wortmannin treatment. Mevastatin showed no significant effect on the parameters γ_RC_ and φ_Po_, and wortmannin showed no significant effect on φ_Po_. The ψ_Εo_ decreased with fumagillin, mevastatin, radicicol, and wortmannin treatment by 17%, 11%, 72%, and 83%, respectively, as shown in Figure 5C. For fumagillin, the decrease in PI_ABS_ is related to the changes in the parameters γ_RC_, φ_Po_, and ψ_Εo_. In other words, fumagillin harmed the antenna size, primary photochemical reaction, and redox reaction after Q_A_, which together significantly reduced the overall photosynthetic activity of PSII. For mevastatin and wortmannin, the decrease in PI_ABS_ was only associated with a significant decrease in the parameter ψ_Εo_. The main mechanism of action of mevastatin and wortmannin is the inhibition of electron transport beyond the Q_A_. As for radicicol, γ_RC_, φ_Po_, and ψ_Εo_ were significantly affected, but ψ_Εo_ was clearly the dominant factor. This suggested that the main factor for the decrease of PI_ABS_ by radicicol as the reduced efficiency of movement in the redox reaction of the electron transport chain. The inhibition of the flow of PSII electrons from Q_A_ to Q_B_ remained a common feature of the mechanism of action of the four mycotoxins.

To summarize, we found that the common and dominant mechanism of action of the four mycotoxins in inhibiting photosynthetic activity was to block electron transfer on the acceptor side of PSII by occupying the Q_B_ site. This further resulted in different degrees of reduction in the number of active PSII RCs and the overall photosynthetic activity of PSII. It has been widely reported that the interruption of electron transport has some negative effects on photosynthesis. For example, reduced electron transport capacity limits the synthesis of ATP and thus inhibits the regeneration of RuBP and the assimilation of CO_2_ [33,34]; the blockage of electron flow leads to an increase in the number of escaped electrons and thus increases the production of dangerous ROS [35]. In addition, we noticed that the multiple negative effects of fumagillin and radicicol on OEC activity, primary photochemical reaction, and antenna size also affected photosynthesis.

### 2.5. Modeling of Four Mycotoxins’ Binding Niche at the Q_B_-Site of the D1 Protein

PSII is an important part of photosynthesis, which provides the initial charge separation to generate high-energy electrons for photosynthetic electron transport. The PSII RC core consists of two highly hydrophobic proteins, D1 and D2, which embed most of the redox active components involved in photosynthetic electron transfer through PSII [19]. The Q_B_ niche on the D1 subunit is a binding target for many known PSII inhibitor herbicides [36], which compete with Q_B_ for the Q_B_ binding site during electron transport, ultimately leading to plant death [37]. The Q_B_-binding pocket is a cavity lined with hydrophobic residues located in the connecting loop between the fourth and fifth transmembrane helices of the D1 protein, which consists of at least 65 amino acids spanning from Phe211 to Leu275 [19]. All these amino acids are highly conserved in representatives of cyanobacteria, algae, and plants, implying that the Q_B_ site is also conserved in oxygenic photosynthetic organisms [38].

To further confirm the binding site of the four mycotoxins in the D1 protein, the crystal structure of the *C. reinhardtii* D1 protein (Protein Data Bank (PDB): 6KAC) was selected to model the position of the four mycotoxins in the Q_B_ site with Discovery Studio (version 2016, BIOVIA, San Diego, CA, USA) (Figure 6 and Figure 7 and Table 1). The proposed molecular model showed that the D1-Phe265 residue was the common binding site of the four mycotoxins in the Q_B_ site, which formed a hydrogen bond with the oxygen atom of four mycotoxins. Interestingly, the hydrogen bond donor of mevastatin was carbonyl oxygen (CO) of the D1-Phe265 residue instead of amino hydrogen (NH). The simulated modeling distances of the hydrogen bonds of fumagillin, mevastatin, radicicol, and wortmannin were 2.53 Å, 2.67 Å, 2.52 Å and 2.39 Å, respectively. The four mycotoxins also established several hydrophobic, Pi-stacking, and hydrogen-bonding interactions with active site residues, as detailed in Table 1. Hydrogen bonding is considered to play a crucial role in the stability of a structure. Van der Waals, hydrophobic, and Pi-stacking interactions are also involved in the stabilization of mycotoxin binding to the Q_B_ site. Compared with the other models, the fumagillin model showed the largest number of bonding interactions with D1. Our previous experimental results also showed that fumagillin had the strongest inhibitory effect on PSII, with the highest affinity in vitro (Figure 2 and Figure 3C). These results suggest that fumagillin may have the highest affinity for the Q_B_ site of the D1 protein among the four mycotoxins.

The standard model of the classical PSII herbicide DCMU docking to the Q_B_ binding site was established to verify the reliability of the binding model. Our model showed that the possible hydrogen binding interaction for DCMU and D1 was formed between the N7 of DCMU and D1-Ser264 Oγ with a 2.37 Å bound distance (Figure 6E and Table 1). The DCMU was completely nestled in the cavity formed by the Q_B_ binding pocket, forming non polar interactions with D1-Leu218, D1-His252, and D1-Leu271 (Figure 7E). Early crystallographic investigations of PSII and studies with resistant mutants support that D1-Ser264 plays a key role in DCMU binding to the Q_B_ site [37,39,40]. Previous studies showed that the amide hydrogen of DCMU may form a hydrogen bond with the hydroxyl oxygen of D1-Ser264. Another weak hydrogen bridge is formed between the carbon group of DCMU and the side chain of D1-His215 [19]. In the herbicide-resistant mutant experiment of *C. reinhardtii*, the residue Ser264 mutation in the D1 protein conferred DCMU resistance [39]. Our modeling is in agreement with the existing data, with high reliability and accuracy.

The surface representations of the four mycotoxins bound to the Q_B_ site showed their positions in the binding pocket (Figure 7). The short alkyl side chain of fumagillin entered inside the cavity, and its long unsaturated carboxyl side chain and cyclohexane ring were partly exposed outside the cavity (Figure 7A). The six-membered oxygen-containing hetero ring of mevastatin entered the Q_B_ binding pocket; other parts lay along the wall of the cavity (Figure 7B). For radicicol, the fifteen-membered hetero ring entered the Q_B_ binding pocket, except for the benzene ring (Figure 7C). The results of the docking model suggested that the four mycotoxins show binding affinity for binding the Q_B_ site. However, DCMU totally enters the cavity formed by the Q_B_ binding pocket (Figure 7E). This may be the reason why the inhibitory activity of the four mycotoxins against PSII was weaker than that of DCMU.

It is well known that the negative effects of weed resistance are usually associated with the long-term and large-scale use of conventional herbicides [41]. PSII inhibitors have been classified as the urea/triazine type or phenol type according to their structural characteristics and modes of inhibition, both of which have their characteristic orientation in D1 proteins. Previous studies have suggested that urea/triazine herbicides prefer to orient toward Ser264, while the preferential binding orientation of phenol herbicides is His215 [36,37]. Due to the proliferation of resistant weeds, the development of herbicides with new target sites has become more urgently required. Natural herbicides have received widespread attention because of their novel structures, unique targets, and environmental friendliness. Some natural PSII inhibitors are tenuazonic acid and patulin, which have unique binding orientation at the Q_B_ site. The docking model of tenuazonic acid to *Arabidopsis* D1 protein showed that tenuazonic acid is preferentially oriented to D1-Gly256 by a hydrogen bond [42]. By docking patulin to the *Ageratina adenophora* D1 protein, patulin was found to bind to the Q_B_ site by forming a hydrogen bond with the His252 residue in the D1 protein [32]. Based on molecular docking models of four mycotoxin bounding to *C. reinhardtii* D1 proteins, we concluded that residue Phe265 is the key residue for the interaction of the four mycotoxins with D1, which is clearly different from the key active site of commercial herbicides and reported natural PSII inhibitors. Therefore, these mycotoxins cannot be directly applied as herbicides but may provide ideas for the development and synthesis of novel herbicides in the future. However, their binding environment needs to be further verified by crystallographic data and mutant experiments.

## 3. Materials and Methods

### 3.1. Plant Materials and Chemicals

The *C. reinhardtii* wild-type strain was obtained from the Freshwater Algae Culture Collection at the Institute of Hydrobiology (FACHB-collection, Chinese Academy of Science, Wuhan, China). *C. reinhardtii* cells were cultured in liquid tris-acetate phosphate media (TAP) under 100 μmol (photons) m^−2^ s^−1^ white light with a 12 h photoperiod at 25 °C. Three-day-old cells in the logarithmic growth phase were collected for further experiments [26].

Fumagillin, mevastatin, radicicol, wortmannin, 3-(3,4-dichlorophenyl)-1,1-dimethylurea (DCMU), dimethyl sulfoxide (DMSO), and other chemical agents were purchased from Sigma-Aldrich (Shanghai, China). Four mycotoxins and DCMU stock solutions were dissolved in 100% DMSO and further diluted with sterile water. The final concentration of DMSO in the working solutions of all chemical agents was less than 1% (*v*/*v*).

### 3.2. Measurement of PSII Oxygen Evolution Rate

The rate of oxygen evolution of PSII was measured using a Clark-type Oxygen Electrode (Hansatech Instruments Ltd., King’s Lynn, UK) according to the method of Guo et al. [43]. After *C. reinhardtii* cells were collected, they were washed and resuspended in buffer A containing 20 mM HEPES (pH 7.5), 350 mM sucrose, and 2.0 mM MgCl_2_ with an A_750_ value of 0.65. Each mycotoxin was added into 2 mL cell suspensions with the indicated concentrations. The samples were incubated for 3 h in the dark at 25 °C. Subsequently, treated cells containing 45 μg of chlorophylls were added into 2 mL of PSII reaction medium (50 mM HEPES-KOH at pH 7.6, 4 mM K_3_Fe(CN)_6_, 5mM NH_4_Cl, and 1 mM p-phenylenediamine). The oxygen evolution rate was determined and recorded during the first three minutes after samples were illuminated with 400 µmol (photons) m^−2^ s^−1^ red light.

### 3.3. Chl a Fluorescence Imaging

Chl *a* fluorescence imaging, F_V_/F_M_, and the ETR of dark-adapted samples were measured using a pulse-modulated Imaging-PAM M-series fluorometer (MAXI-version, Heinz Walz GmbH, Effeltrich, Germany) [43]. Cells were resuspended with buffer A. We added 200 μL of cell suspension with the indicated concentrations of fumagillin, mevastatin, radicicol, wortmannin, or 1% DMSO (mock) to a 96-well black microtiter plate. The cell suspensions were incubated for 2.5 h under light (100 μmol m^−2^ s^−1^) at 25 °C. Then, samples were dark-adapted for 30 min under the imaging system camera. For monitoring fluorescence imaging, the measuring light, actinic light, and saturation pulse light were set to 0.25, 110, and 6000 μmol (photons) m^−2^ s^−1^, respectively.

### 3.4. Chl a Fluorescence Rise Kinetics OJIP and JIP-Test

The Chl *a* fluorescence rise kinetics OJIP were measured with a Plant Efficiency Analyzer (Hansatech Instruments Ltd., King’s Lynn, UK) [27]. We incubated 1 mL of cell suspension in buffer A with 1% DMSO (mock), 50 µM DCMU, 50 µM fumagillin, 200 µM mevastatin, 500 µM radicicol, or 500 µM wortmannin for 2.5 h under 100 (photons) µmol m^−2^ s^−1^ white light at 25 °C. The cells were collected and resuspended in 20 µL of buffer A. After 30 min dark adaptation, 20 µL of the sample was filtered onto a glass microfiber filter (diameter 25 mm, GF/C, Whatman, Kent, UK) and clamped with a leaf clip. Samples were illuminated with continuous red light (650 nm peak wavelength, 3500 µmol (photons) m^−2^ s^−1^ maximum light intensity). The raw data were transferred to a computer using Handy PEA software (version 1.30, Hansatech Instruments Ltd., Norfolk, UK). The experiment was repeated three times with at least 15 repetitions. The detailed parameters and definitions re listed in Table 2, according to Strasser et al. [21] and Chen et al. [27].

### 3.5. Modeling of Four Mycotoxins in the Q_B_ Binding Site

The crystal structure information of the D1 protein of *C. reinhardtii* was obtained from the Protein Data Bank (https://rcsb.org, accessed on 17 January 2023, PDB code: 6KAC; the resolution: 2.70 Å). Its dimeric structure was optimized by CHARMm force field using Discovery Studio (version 2016, BIOVIA, San Diego, CA, USA). The structures of DCMU and the four mycotoxins as ligands were constructed using ChemDraw 18.0 (Cambridge Soft, MA, USA). The ligand structures were energetically minimized using the MM2 energy minimizations tool in Chem3D Pro 14.0 (Cambridge Soft, MA, USA). The possible binding site of ligands docking was set to the Q_B_ binding site in the D1 crystal structure of *C. reinhardtii* (PDB: 6KAC). The molecular docking was performed with CDocker in Discovery Studio. For the setting of the docking parameters, the Top Hits was set to 10, and the Pose Cluster Radius was set to 0.5 Å. We used the default values for the other parameters.

### 3.6. Statistical Analysis

One-way ANOVA was carried out, and means were separated by Duncan’s LSD at 95% using SPSS Statistics 20.0 (IBM, CA, USA).

## 4. Conclusions

In conclusion, we showed that the mycotoxins fumagillin, mevastatin, radicicol, and wortmannin exhibit multiple negative effects on the photosynthetic process of *C. reinhardtii* cells. These mycotoxins disrupt PSII RCs by occupying the Q_B_ site, then block the electron transport flow from Q_A_ to Q_B_, and finally inhibit photosynthetic activity. The four mycotoxins form a hydrogen bond with the Q_B_ site via the D1-Phe265 residue, which is different from classical PSII herbicides. The structures of these mycotoxins may provide ideas for the discovery of novel derivatives with stronger herbicide activity. However, the more precise binding conditions of the mycotoxins need to be further explored.

## Figures and Tables

**Figure 1 plants-12-00665-f001:**
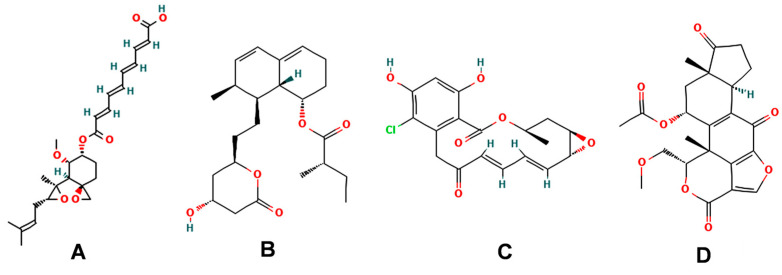
Chemical structure of four mycotoxins: (**A**) fumagillin (C_26_H_34_O_7_, MW: 458.5), (**B**) mevastatin (C_23_H_34_O_5_, MW: 390.5), (**C**) radicicol (C_18_H_17_ClO_6_, MW: 364.8), and (**D**) wortmannin (C_23_H_24_O_8_, MW: 428.4).

**Figure 2 plants-12-00665-f002:**
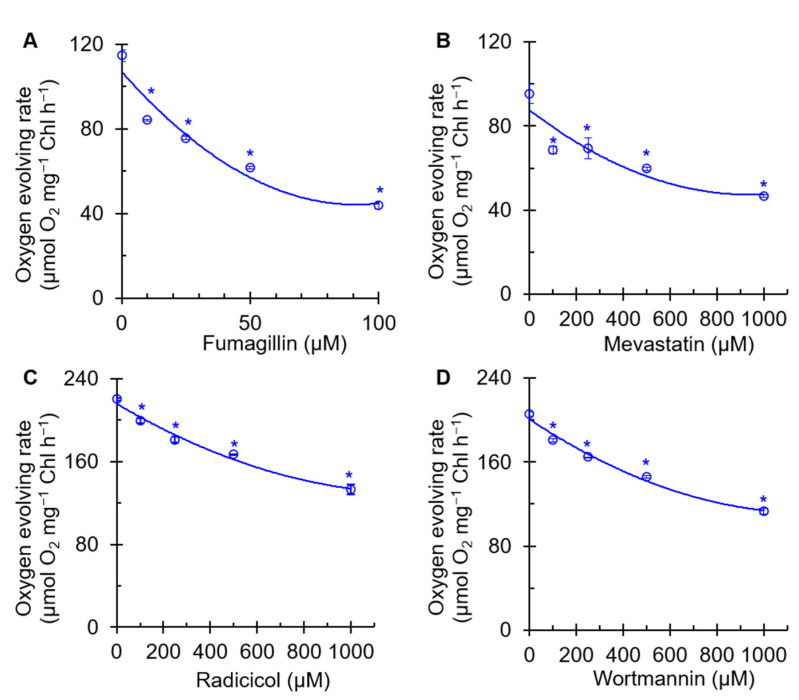
Effects of four mycotoxins, fumagillin (**A**), mevastatin (**B**), radicicol (**C**), and wortmannin (**D**), on oxygen evolving rate of *C. reinhardtii* cells. H_2_O and p-phenylenediamine were the electron donor and acceptor, respectively. Data shown are mean values ± SE of three biological replicates. * indicates significance at *p* < 0.05.

**Figure 3 plants-12-00665-f003:**
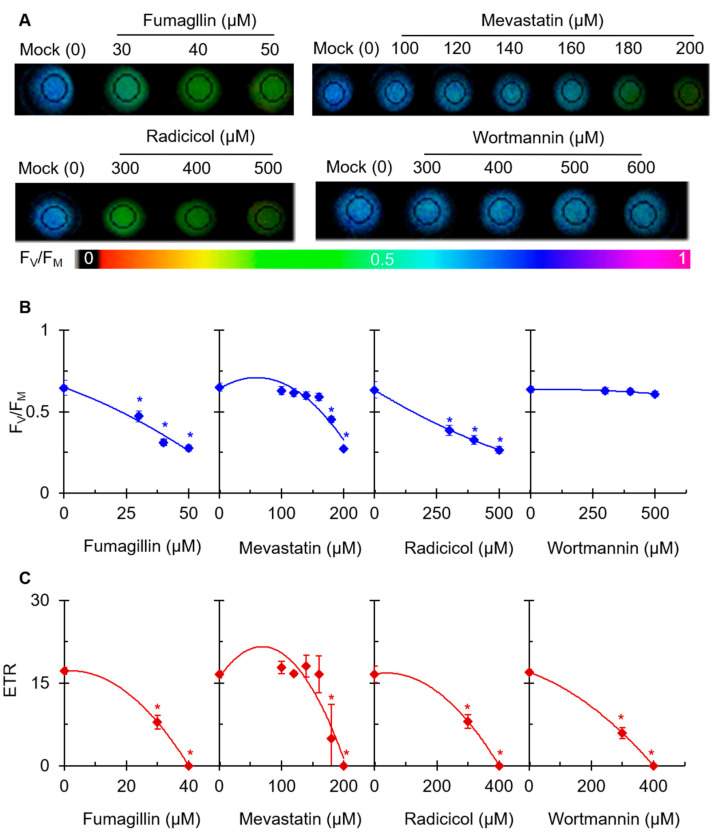
Effects of four mycotoxins on Chl fluorescence of *C. reinhardtii* cells. Cells were incubated for 3 h with different concentrations of fumagillin (30, 40, and 50 μM), mevastatin (100, 120, 140, 160, 180, and 200 μM), radicicol (300, 400, and 500 μM), and wortmannin (300, 400, 500, and 600 μM). (**A**) Color fluorescence images of the maximum quantum yield of PSII (F_V_/F_M_). (**B**) F_V_/F_M_ value. (**C**) The electron transport rate (ETR). Fluorescence images are indicated by the color code in the order of black (0) through red, orange, yellow, green, blue, and violet to purple (1). Each value is the average ± SE of three biological replicates. * indicates significance at *p* < 0.05.

**Figure 4 plants-12-00665-f004:**
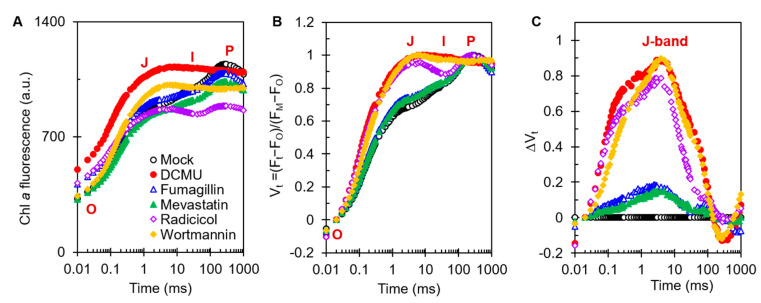
Effects of four mycotoxins and DCMU on Chl *a* fluorescence rise kinetics of *C. reinhardtii* cells. Cells after 3 h of treatment with 0.1 % DMSO (mock), 50 μM DCMU, 50 μM fumagillin, 200 μM mevastatin, 500 μM radicicol, or 500 μM wortmannin. (**A**) Raw fluorescence rise OJIP curves of mock-, DCMU-, fumagillin-, mevastatin-, radicicol-, and wortmannin-treated *C. reinhardtii* cells. (**B**) Chl *a* fluorescence rise kinetics normalized by F_O_ and F_M_ as V_t_ = (F_t_ − F_O_)/(F_M_ − F_O_) versus logarithmic timescale. (**C**) The difference kinetics ∆V_t_ = V_t(treatment)_ − V_t(mock)_ versus logarithmic timescale. Each curve is the average of 30 measurements.

**Figure 5 plants-12-00665-f005:**
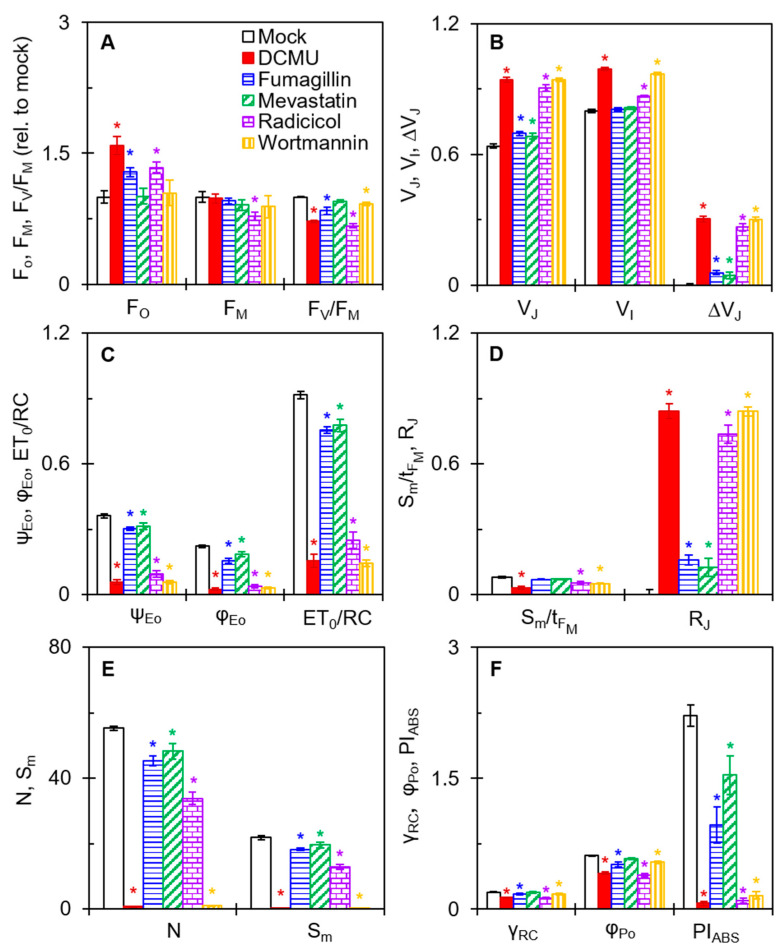
Effects of four mycotoxins on the selected JIP test parameters of *C. reinhardtii* cells. (**A**) The relative values of F_O_, F_M_, and F_V_/F_M_ after mycotoxins treatment compared with mock. (**B**) The values of V_J_, V_I_, and ΔV_J_. (**C**) The values of ψ_Eo_, φ_Eo_, and ET_0_/RC. (**D**) The values of S_m_/t_FM_ and R_J_. (**E**) The values of N and S_m_. (**F**) The values of γ_RC_, φ_Po_, and PI_ABS_. Each parameter is the average of 30 measurements. * indicates significance at *p* < 0.05.

**Figure 6 plants-12-00665-f006:**
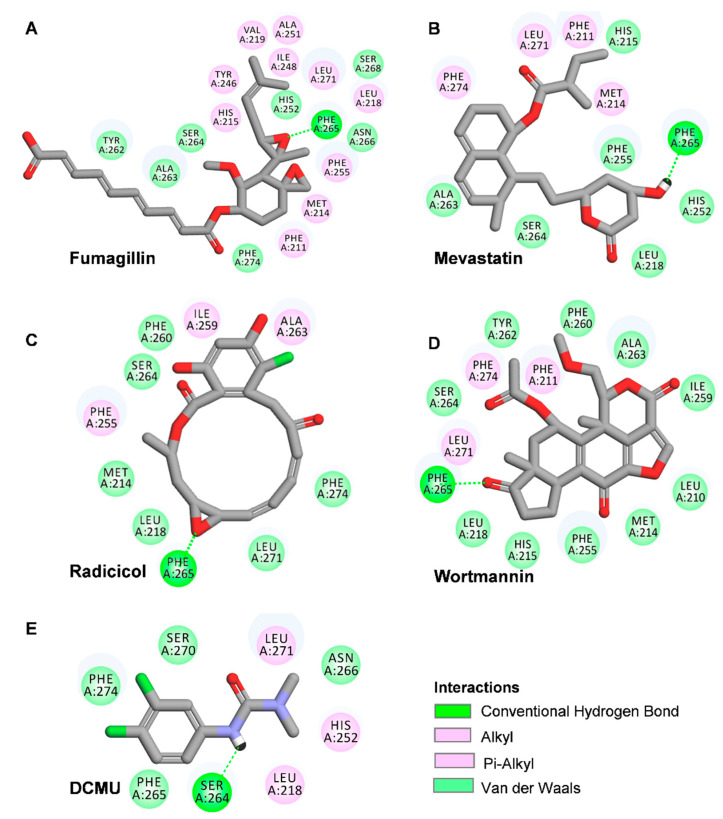
Hydrogen-bonding interactions for fumagillin (**A**), mevastatin (**B**), radicicol (**C**), wortmannin (**D**). and DCMU (**E**) in the Q_B_ binding site of D1 protein of *C. reinhardtii* (6KAC). Here, carbon atoms are shown in grey, nitrogen atoms in blue, oxygen in red, chlorine atoms in green, and hydrogen atoms in white. Possible hydrogen bonds are indicated by a dashed line.

**Figure 7 plants-12-00665-f007:**
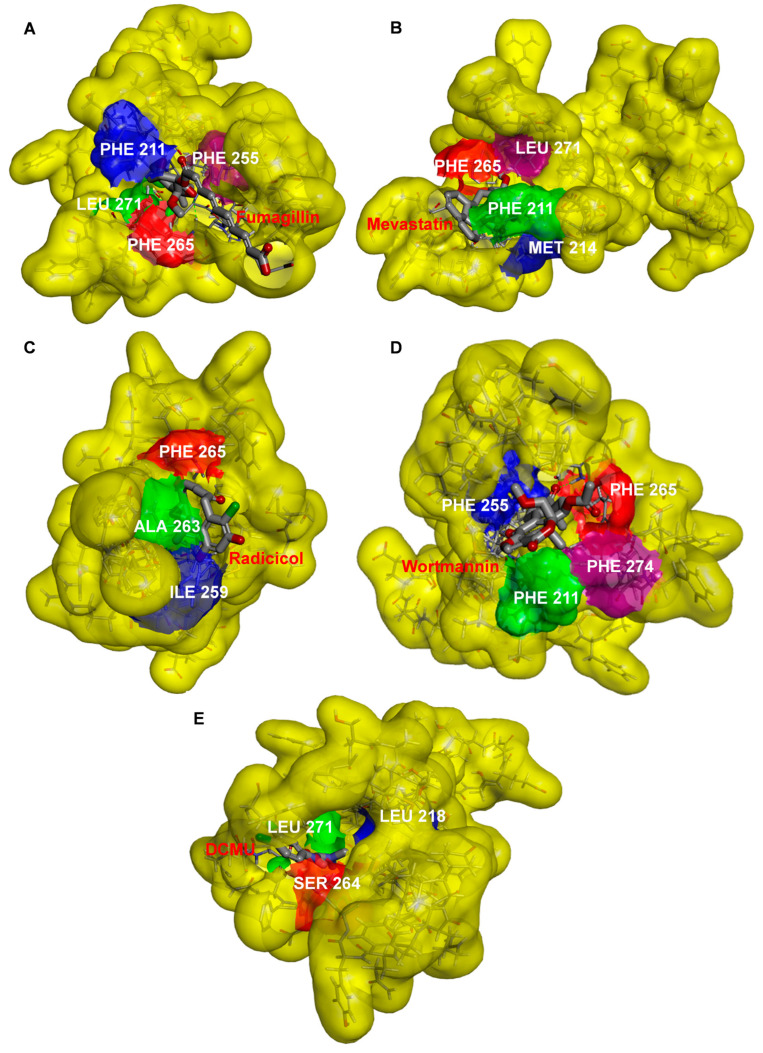
Docked poses of fumagillin (**A**), mevastatin (**B**), radicicol (**C**), wortmannin (**D**), and DCMU (**E**) inside the Q_B_ binding site of D1 protein of *C. reinhardtii* (6KAC).

**Table 1 plants-12-00665-t001:** Possible binding interactions for DCMU, fumagillin (Fum), mevastatin (Mev), radicicol (Rad), and wortmannin (Wor) to the D1 protein of *C. reinhardtii.*

Compound	Molecular Formula	Chemical Structure	Donor	Acceptor	Interactions	Bound Distance (Å)
DCMU	C_9_H_10_Cl_2_N_2_O	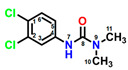	D1-Leu 218	DCMU C10	Alkyl Hydrophobic	3.15
			D1-His 252	DCMU C11	Alkyl Hydrophobic	3.22
			D1-Ser 264 O*γ*	DCMU NH	Hydrogen Bond	2.37
			D1-Leu 271	DCMU C11	Alkyl Hydrophobic	3.17
Fumagillin	C_26_H_34_O_7_	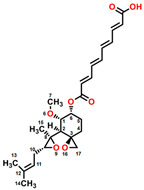	D1-Phe 211	Fum C17	Pi-Hydrophobic	3.52
			D1-Met 214	Fum C17	Alkyl Hydrophobic	3.48
			D1-His 215	Fum C7	Alkyl Hydrophobic	3.63
			D1-Leu 218	Fum C13	Alkyl Hydrophobic	3.67
			D1-Val 219	Fum C13	Alkyl Hydrophobic	3.18
			D1-Tyr 246	Fum C13	Alkyl Hydrophobic	3.23
			D1-Ile 248	Fum C12	Alkyl Hydrophobic	3.20
			D1-Ala 251	Fum C12	Alkyl Hydrophobic	3.49
			D1-Phe 255	Fum C15	Pi Hydrophobic	3.64
			D1-Phe 265 NH	Fum O9	Hydrogen Bond	2.53
			D1-Leu 271	Fum C12	Alkyl Hydrophobic	3.36
Mevastatin	C_23_H_34_O_5_	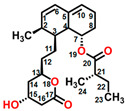	D1-Phe 211	Mev C23	Pi Hydrophobic	3.21
			D1- Met 214	Mev C9	Alkyl Hydrophobic	3.18
			D1-Phe 265 CO	Mev C15-OH	Hydrogen Bond	2.67
			D1-Leu 271	Mev C23	Alkyl Hydrophobic	3.55
			D1-Phe 274	Mev C24	Pi Hydrophobic	3.46
Radicicol	C_18_H_17_ClO_6_	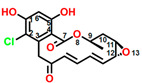	D1-Phe 255	Rad C9-CH_3_	Pi Hydrophobic	3.31
			D1-Ile 259	Rad Ph	Pi Hydrophobic	3.64
			D1-Ala 263	Rad Ph	Pi Hydrophobic	3.57
			D1-Phe 265 NH	Rad O13	Hydrogen Bond	2.52
Wortmannin	C_23_H_24_O_8_	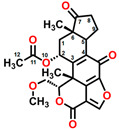	D1-Phe 211	Wor C12	Pi Hydrophobic	3.51
			D1-Phe 265 NH	Wor O7	Hydrogen Bond	2.39
			D1-Leu 271	Wor C6-CH_3_	Alkyl Hydrophobic	3.28
			D1-Phe 274	Wor C12	Pi Hydrophobic	3.37

**Table 2 plants-12-00665-t002:** Formulae and explanation of technical data of the OJIP curves and the selected JIP test parameters used in this study.

Technical Fluorescence Parameter	
F_t_	fluorescence at time t after onset of actinic illumination
F_O_ ≅ F_20μs_	minimal fluorescence, when all PSII RCs are open
F_K_ ≡ F_300μs_	fluorescence intensity at the K step (300 μs) of OJIP
F_J_ ≡ F_2ms_	fluorescence intensity at the J step (2 ms) of OJIP
F_I_ ≡ F_30ms_	fluorescence intensity at the I step (30 ms) of OJIP
F_P_ (=F_M_)	maximal recorded fluorescence intensity, at the peak P of OJIP
F_v_ ≡ F_t_ − F_O_	variable fluorescence at time t
F_V_ ≡ F_M_ − F_O_	maximal variable fluorescence
t_FM_ Area	time (in ms) to reach the maximal fluorescence intensity F_M_ total complementary area between the fluorescence induction curve and F = F_M_
V_t_ ≡ (F_t_ − F_O_)/(F_M_ − F_O_)	relative variable fluorescence at time t
V_J_ = (F_J_ − F_O_)/(F_M_ − F_O_) V_I_ = (F_I_ − F_O_)/(F_M_ − F_O_) M_0_ ≡ 4(F_270μs_ − F_O_)/(F_M_ − F_O_) S_m_ ≡ Area/(F_M_ − F_O_)	relative variable fluorescence at the J step relative variable fluorescence at the I step transient normalized on the maximal variable fluorescence F_V_ normalized total complementary area above the O-J-I-P transient (reflecting multiple-turnover Q_A_ reduction events)
N ≡ S_m_/S_s_ = S_m_·M_0_/V_J_	turnover number: number of Q_A_ reduction events between time 0 and t_FM_
Q**uantum efficiencies or flux ratios**	
φ_Po_ = PHI(P_0_) = TR_0_/ABS = 1 − F_O_/F_M_	maximum quantum yield for primary photochemistry
ψ_Eo_ = PSI_0_ = ET_0_/TR_0_ = (1 − V_J_)	probability that an electron moves further than Q_A_^−^
φ_Eo_ = PHI(E_0_) = ET_0_/ABS = (1 − F_O_/F_M_) (1 − V_J_)	quantum yield for electron transport (ET)
γ_RC_ = Chl_RC_/Chl_total_ = RC/(ABS + RC)	probability that a PSII Chl molecule functions as RC
**Specific energy fluxes**	
ET_0_/RC = M_0_·(1/V_J_)·(1 − V_J_)	electron transport flux per RC (at t = 0)
**Density of RCs**	
S_m_/t_FM_ = [RC_open_/(RC_close_ + RC_open_)] av R_J_ = [ψ_Eo(mock)_ − ψ_Eo(treatment)_]/ ψ_Eo(mock)_	average fraction of open RCs of PSII in the time span between 0 to t_FM_ number of PSII RCs with Q_B_-site filled by PSII inhibitor
**Performance indexes**	
${\mathrm{PI}}_{\mathrm{ABS}}$ ≡ $\frac{\gamma_{\mathrm{RC}}}{1-\gamma_{\mathrm{RC}}}\cdot \frac{\phi_{\mathrm{Po}}}{1-\phi_{\mathrm{Po}}}\cdot \frac{\psi_{\mathrm{Eo}}}{1-\psi_{\mathrm{Eo}}}$	performance index (potential) for energy conservation from photons absorbed by PSII to the reduction of intersystem electron acceptors

Subscript “0” (or “o” when written after another subscript) indicates that the parameter refers to the onset of illumination, when all RCs are assumed to be open.

## Data Availability

Not applicable.

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
