# Peer review of "Effects of Mycotoxin Fumagillin, Mevastatin, Radicicol, and Wortmannin on Photosynthesis of *Chlamydomonas reinhardtii"

_plants, 2023, doi:10.3390/plants12030665_

Round 1
Reviewer 1 Report
Dear authors,
1. check Mevastatin line 72
2. check figure 2 title
3. check statistics: ANOVA was not employed as stated
4. concentration choice? – Did the effects depend on the concentration? Please, add the information.
5. Mycotoxins as poisons is a bad choice for direct application of pesticides. Could you stress that these compounds will be just a model for the synthesis? Please, add the information, if appropriate.
6. Did you assay the algal growth inhibition or phytotoxicity to plants treated with the 4 mycotoxins used in this study? Is there correlation between ChlA inhibition and the phytotoxicity? Please, add the information.
7. In your this and previous research many compounds (different mycotoxins) affected D1 protein. What a mechanism or explanation? Please, add the information or your opinion.
Reviewer 2 Report
Dear authors,
The manuscript entitled "Effects of Mycotoxin Fumagillin, Mevastatin, Radicicol and Wortmannin on Photosynthesis of Chlamydomonas reinhardtii" adds valuable information regarding the effects of mycotoxins on PSII, suggesting the development of novel herbicides.
The introduction is well-written and organized and it was easy to read. I really enjoyed reading it. The material and methods are also well described which explains all the methodology adopted by the authors to conduct the whole experiment. The results and discussion also present timely information and adequate references.
I recommend the authors to check all figures' citations that must be added to the text before the figure appears.
There are some minor corrections that I have added in the PDF version.
Kind Regards

Author Response
Thank you very much. We are truly appreciative of the reviewer’s enthusiasm and encouraging comments, which were very helpful in revising the manuscript.
We have made changes based on your friendly suggestions and highlighted them with tracking changes.
Reviewer 3 Report
The manuscript “Effects of mycotoxin fumagillin, mevastatin, radicicol and wortmannin on photosynthesis of Chlamydomonas reinhardtii” gives very interesting study that investigated impact of four mycotoxins on photosynthesis. In the study it is found that mycotoxins affect photosynthesis, and therefore their chemical structure can provide ideas for discovering new herbicides.
In the study state-of-the-art methods were used, and several endpoints were assessed. Clear conclusions based on obtained results were made. There are only some minor comments that authors could introduce in the manuscript.
Minor comments
In the manuscript it is not explained why Chlamydomonas reinhardtii is used for experiments.
Check Figure 2 caption. “Chemical structure of four mycotoxins” should be omitted.
Author Response
RE: We thank the reviewer for his general appreciation. As explained below, we have revised the manuscript in light of these helpful comments. We would like to also thank this reviewer for his effort in the whole review process.
Minor comments
In the manuscript it is not explained why Chlamydomonas reinhardtii is used for experiments.
RE: Thanks. Line 108-111 in track changes version.
Check Figure 2 caption. “Chemical structure of four mycotoxins” should be omitted.
RE: Thanks. Line 134 in track changes version.